# Nutritional Composition and Biological Properties of Sixteen Edible Mushroom Species

**Maria Dimopoulou [1], Alexandros Kolonas [1], Stamatis Mourtakos [2], Odysseas Androutsos [3] and Olga Gortzi [1,\*]**

[1] Department of Agriculture Crop Production and Rural Environment, School of Agricultural Sciences, University of Thessaly, 38446 Volos, Greece

[2] Department of Nutrition and Dietetics, School of Health Science and Education, Harokopio University of Athens, 17671 Athens, Greece

[3] Department of Nutrition & Dietetics, School of Physical Education, Sport Science and Dietetics, University of Thessaly, 42100 Trikala, Greece

[\*] Correspondence: olgagortzi@uth.gr; Tel.: +30-2421093289

**Abstract:** Mushrooms are considered to be functional foods with high nutritional, culinary, and pharmacological values, and there has been an increase in their consumption, both through the diet and in the form of dietary supplements. The present study aimed to briefly review the nutritional composition and biological properties of sixteen mushroom species, as well as to compare the mushrooms' proximate composition to the analyses conducted at the University of Thessaly, Greece, in cooperation with the Natural History Museum of Meteora and Mushroom Museum. The macronutrient profile of each mushroom was analyzed according to the methods described in the Association of Official Analytical Chemists International, at the School of Agricultural Sciences of the University of Thessaly. The protein content of the mushrooms was found to range between 13.8 g/100 g and 38.5 g/100 g, carbohydrate content ranged between 32 g/100 g and 61.4 g/100 g, and fat content ranged between 0.4 g/100 g and 5.9 g/100 g. Additionally, a serving of 100 g of most species of mushrooms covers 15 to 30% of the daily recommendation of vitamins and trace elements. Based on their compositions, mushrooms were shown to constitute excellent food sources from a nutritional point of view, containing high amounts of dietary fiber and protein, low fat, and reasonable sources of phosphorus, although they were shown to be poor in vitamin C.

**Keywords:** mushroom; nutritional value; edible fungi; proximate composition

## 1. Introduction

Mushrooms are considered to be functional foods with high nutritional, culinary, and pharmacological values [1]. Mushrooms have been recognized for the aforementioned values since ancient times. Ancient Greek warriors consumed mushrooms to increase their strength in battles, the Romans considered mushrooms as the food of the Gods, the Egyptians thought that mushrooms were a gift from Osiris, and the Chinese used mushrooms in medicine and believed that they promote youth and health [2].

Mushrooms represent a global market of USD 63 billion, with 54% of this market being cultivated edible mushrooms, 38% medicinal mushrooms, and 8% wild mushrooms [3]. Worldwide mushroom production reached 10,378,163 metric tons in 2016 and the average per-capita consumption has grown considerably in recent years [4]. Specifically, global consumption of mushrooms has increased from 1 to 4.7 kg of cultivated edible mushrooms per capita from 1997 to 2013 [3]. China is currently the leading producer of cultivated edible mushrooms worldwide accounting for approximately 73 percent of the world's total mushroom production [2,5]. The most cultivated mushrooms in the world are *Agaricus bisporus*, *Lentinula edodes*, *Pleurotus* spp., *Auricularia auricula-judae*, *Volvariella volvacea*, and *Flammulina velutipes* [6]. On the other hand, the most famous wild mushrooms are *Boletus edulis*, *Cantharellus* spp., *Craterellus cornucopioides*, *Morchella* spp., and *Marasmius oreades* [5].

Mushrooms form the three groups of fungi, Zygomycetes, Ascomycetes and Basidiomycetes. All three categories can grow both above the soil (epigeous) and below the soil (hypogeous) [7]. Fungi lack chlorophyll and the ability to utilize photosynthesis and thus are distinct from the plant kingdoms [4]. Most mushroom species' cultivation depends highly on environmental factors, such as temperature, air, and humidity, in order to grow optimally [4]. Traditionally, mushrooms are cultivated using outdoor log cultures, which have been used in China for over one thousand years. Nowadays, indoor cultivation in "artificial logs" is commonly used, which are plastic bags filled with nutrient-complemented sawdust-based substrates. The sawdust is held together like glue and when the bag is colonized it is unpacked to allow fruiting. Another similar cultivation technique is the column cultures, which are long plastic bags that are hung from the ceiling and once the mycelium colonizes the bags, holes are punched to allow mushroom fruiting [3].

Mushrooms are of high nutritional value due to the various nutrients they contain, such as protein, including essential amino acids, essential fatty acids, carbohydrates, dietary fiber, vitamins, and minerals [8]. In addition, there is a plethora of bioactive compounds in mushrooms, such as polysaccharides, polyphenols, terpenoids, lectins, alkaloids, sterols, glucoproteins, ergosterols, sesquiterpenes, and lactones [7]. However, the contents of these bioactive compounds may vary significantly depending on various factors, such as strain, substrate, cultivation, storage conditions, and processing [7]. Due to the aforementioned bioactive compounds, mushrooms exhibit a wide array of health-promoting properties including antioxidant, anti-cancer, immunomodulatory, anti-diabetic, neuroprotective, anti-hypertensive, hepatoprotective, anti-fungal, anti-microbial, anti-viral and anti-bacterial properties [9].

However, until recently the scientific understanding of mushrooms' medicinal properties has been primarily empirical [10]. There has been an increasing interest in advancing the health properties and pharmacological activities resulting in the growth of clinical research. Within this framework, much of the traditional knowledge on the issue is being documented and validated [11]. There is now an interdisciplinary field of science that studies medicinal mushrooms (MMs), accompanied by an increasing number of emerging human studies. These scientific advances coupled with industry technological developments have resulted in some mushrooms being now regarded as a distinct class of drugs called "mushroom pharmaceuticals" [12].

Throughout Europe, as in Greece as well, there are about 4000–5000 species of mushrooms [13] and although hundreds of them are edible, only sixteen species reach our tables. Mushrooms are basic ingredients for many traditional dishes more often due to the popularity of the 14 awards at the Mediterranean Taste Awards 2021 (MTA 2021) and Olymp Awards that the new biofunctional products received.

There is witnessed an increase in the use of medicinal mushrooms (MMs) in the form of extracts in nutraceuticals and other functional foods and health products. There are several other places in Greece where mushrooms thrive, including Valia Calda, Zagori, Pelion, Kastoria, Metsovo, and Kalambaka. However, it is of prime consideration that mushrooms' properties, mechanisms of action, and potential efficacies can be affected by many variables, including climate, location, cultivation, processing, and extraction techniques [11]. The most common edible mushrooms in Greece are *Agaricus crocodilinus, Amanita caesarea, Boletus aereus, Boletus reticulatus, Cantharellus subpruinosus*, (which is also called *Cantharellus pallens*) *Coprinus comatus, Craterellus cornucopioides, Craterellus lutescens, Cyclocybe cylindracea, Hydnum spp., Lactarius deliciosus, Lactarius salmonicolor, Marasmius oreades, Macrolepiota procera, Morchella spp., Pleurotus ostreatus, Russula aurea, Russula virescens*, and *Tuber aestivum.*

The current paper aims to briefly review sixteen mushrooms species (*Agaricus bisporus, Agaricus blazei, Amanita caesarea, Boletus edulis, Cantharellus cibarius, Coprinus comatus, Cordyceps militaris, Craterellus cornucopioides, Craterellus lutescens, Ganoderma lucidum, Grifola frondosa, Hericium erinaceus, Lentinula edodes, Marasmius oreades, Morchella elata, and Pleurotus citrinopileatus*) regarding their chemical and nutritional composition, as well as their medic-

inal and biological properties. In order to search for the information, a literature search was carried out in the databases: PubMed, Science Direct, and Google Scholar database. The search also included articles which were bibliographic references to the articles that were studied.

## 2. Materials and Methods

A sample of dehydrated mushrooms of each variety was obtained from the cooperation with the Meteora Museum. To allow greater extraction of its components, the mushroom was mashed up in a Willey type (Model ET-648, Tecnal Brand mill). The physical and chemical analyses were performed at the "Food InnovaLab" of the Department of Food Technology of Technological Educational Institute of Thessaly, Karditsa, Greece and at the Food Safety and Quality Control Laboratory of the School of Agricultural Sciences, of the University of Thessaly, Volos, Greece.

## 3. Chemical Characterization

The whole analysis, in duplicate, followed the official methods established by MAPA, by the Association of Official Analytical Chemists (AOAC) [14]. Moisture analysis was performed using a kiln at 105 °C $\pm$ 3 °C for 24 h and total ash by means of sample calcination in a muffle furnace at 550 °C for 12 h. The Kjeldahl method was utilized for protein determination, using a 6.25 correction factor. Sample fat content was detected by continuous "Soxhlet" device type extraction. Determination of total dietary fiber was based on sequential enzymatic digestion of the dried mushroom sample with alpha-amylase thermostable; protease and amyloglucosidase. The determination of carbohydrates was calculated by the difference, using rates obtained by moisture analysis, fixed mineral residue, proteins, and lipids.

## 4. Sixteen Edible Mushrooms of Greece

### 4.1. Agaricus bisporus

*Agaricus bisporus (A. bisporus),* commonly known as the button mushroom, is an edible species of the family Agaricaceae, accounts for 15% of the total mushroom production worldwide, and is cultivated throughout Europe and North America [5]. Dry matter (DM) of cultivated and wild-growing *Agaricus* mushrooms varies and ranges between 83 and 285 g/kg [15]. Proximate compositions of A. bisporus have described moisture (91–92 g/100 g DM), ash (0.9–1 g/100 g DM), energy (29–31 kcal/100 g DM), protein (29.14 g/100 g DM), carbohydrate (51.05 g/100 g DM), and fat (1.56 g/100 g DM) [2]. The main amino acids found in *A. bisporus* are aspartic acid, histidine, glutamic acid, lysine, and serine [4]. The lipid content of *A. bisporus* is low with the main fatty acids being linoleic acid (61.8–67.3% of the total fatty acid content) and palmitic acid (12.7–14.7% of the total fatty acid content) [16]. Furthermore, nutrients such as phenolic compounds have been identified and may be responsible for the antioxidant properties of several mushroom species [17]. Specifically, *A. bisporus* contains *trans*-cinnamic acid (9.4 mg/100 g DM) and chlorogenic acid (5.8 mg/100 g DM) [17]. The main polysaccharides are D-glucans, while chitin and other heteropolysaccharides are found in small amounts [4]. When it comes to the analyses of the University of Thessaly, the macronutrient content was found to be similar (protein 25.1 g/100 g DM, carbohydrate 52.7 g/100 g DM, and fat 0.9 g/100 g DM). Furthermore, these mushrooms are considered to be good sources of vitamins B1, B2, B3, niacin, folate, B12, D2, and ergosterol (biological precursor to vitamin D2), although the content of these vitamins varies depending on growing conditions [18]. Regarding mineral content, *A. bisporus* has been described as a good source of K, Fe, Zn, Cu, Na, Se, Co, and Mn [19]. In addition, *A. bisporus* has been reported to possess antioxidant, anti-diabetic, and antibacterial properties, possibly due to its polysaccharide and phenolic content [18]. Specifically, a study evaluated the antioxidant ability of *A. bisporus* polysaccharide extracts using the 2,2-diphenyl-1-picrylhydrazyl (DPPH) assay and the results showed that at 250 µg/mL the extract exhibited 86.1% free radical scavenging activity, which was significantly higher

($p < 0.01$) than BHT (83%) [20]. In addition, it has been reported that oral administration of high doses of *A. bisporus* extract could lead to decreased severity of streptozotocin-induced diabetes in Sprague-Dawley rats. The rats were fed *A. bisporus* powder (200 mg/kg of body weight) for three weeks, which led to significantly reduced triglyceride concentration (39.1%), plasma glucose concentration (24.7%), aspartate aminotransferase (15.7%), and alanine aminotransferase (11.7%) [21].

### 4.2. Agaricus blazei

*Agaricus blazei* (*A. blazei*), also known as sun mushroom, belongs to the family Agaricaceae and is widely used both as an edible mushroom and for medicinal purposes due to its contents in bioactive compounds, such as phenolic compounds and polysaccharides [22,23]. The chemical composition of dried powder of *Agaricus blazei* has been studied and it has been found that it contains 379.24 kcal/100 g DM, 59.42 g/100 g DM carbohydrates, 31.29 g/100 g DM proteins, 1.82 g/100 g DM fat, and 7.47 g/100 g DM ash [22]. In the analyses of the University of Thessaly, it was found to contain 335 kcal/100 g DM, 48.4 g/100 g DM carbohydrates, 28.6 g/100 g DM proteins, 1.6 g/100 g DM fat, and 8.4 g/100 g DM ash. When it comes to the fatty acids distribution, 73.58% comes from PUFA, 24.39% from saturated fatty acids (SFA), and 2.03% from monounsaturated fatty acids (MUFA), while the main fatty acids are linoleic, palmitic, stearic, and oleic acids [22]. Regarding the phenolic compounds composition, *A. blazei* has been shown to contain p-hydroxybenzoic acid, *trans*-p-coumaric acid, and cinnamic acid [22]. Furthermore, several studies have been conducted to evaluate the medicinal and biological properties of *A. blazei* and it has been found that this mushroom possesses antioxidant [22], anti-viral [24], anti-diabetic [25], immunomodulatory [26], and anti-cancer properties [27]. Regarding its hypoglycemic effects, *A. blazei* has been shown to significantly suppress the increases in fasting plasma glucose and hemoglobin A1c levels, reduce superoxide production from leukocytes, as well as improve the mass of pancreatic β-cells in streptozotocin-induced diabetic rats [25]. Another in vivo study regarding *A. blazei* immune responses in leukemia mice showed that *A. blazei* promoted natural killer cell activity and phagocytosis by macrophage/monocytes and enhanced cytokines such as interleukin (IL)-1β, IL-6, and interferon-γ levels [26]. Lastly, the health claims proposed for *Agaricus blazei* dietary supplements are: "Helps the natural defences contributing to a normal immune response," "helps the body to resist biologic insults," "supports the immune system," and "is rich in beta glucans that contributes to the immune activity" [28].

### 4.3. Amanita caesarea

*Amanita caesarea (A. caesarea)* belongs to the Amanitaceae family and can be found in Yunnan province in China at an elevation of 3800 m [29]. It is commonly known as Caesar's mushroom and it has a characteristic orange cap, yellow gills, and stem [30]. *Amanita caesarea* from Thessaly contains 24% protein, 31.9% carbohydrates, and 5.6% fat, of which 28.6% are unsaturated fatty acids according to the analysis of the University's researchers but from west Macedonia and Epirus contains 34.77% protein, 55.63% carbohydrates, and 3.5% fat [31]. In addition, *Amanita caesarea* contains several phenolic compounds, such as catechin (32.5 mg/g), ferulic acid (7 mg/g), p-coumaric acid (6 mg/g), and cinnamic acid (6.2 mg/g) owing to its antioxidant properties [30]. Furthermore, 37 fatty acids have been identified in *Amanita caesarea*, with oleic acid (58%) being the most prominent [30]. The polysaccharides of *A. caesarea* have been studied for their neuro-protective effects, especially for Alzheimer's disease [32]. In an Alzheimer's disease mouse model, the polysaccharides of *A. caesarea* lead to suppressed deposition of the β-amyloid peptide (the main cause of Alzheimer's disease is a failure to clear the β-amyloid peptide from brain tissue) [33] in the brain and ameliorated oxidative stress, while also improving the functioning of the central cholinergic system, making *A. caesarea's* polysaccharides a promising candidate for the treatment of Alzheimer's disease [32]. Another study, regarding *A. caesarea* polysaccharides and Alzheimer's disease, found that these polysaccharides possess neuroprotective effects

and mitigated Alzheimer's disease-like symptoms in mice through their ability to prevent the development of endoplasmic reticulum stress and oxidative stress [34].

### 4.4. Boletus edulis

*Boletus edulis* (*B. edulis*) belongs to the family Boletaceae, is known as king bolete, and is a symbiotic ectomycorrhizal species with a tubular hymenophore and is native to Europe [35]. It is a highly appreciated mushroom species due to its nutritional value, as well as its exceptional flavor [36]. Fresh *B. edulis* has been shown to contain 45 kcal/100 g, 88.84 g/100 g fresh weight moisture, 2.27 g/100 g proteins, 7.37 g/100 g carbohydrates, 0.87 g/100 g fat, and 0.66 g/100 g ash [36]. Furthermore, *B. edulis* possesses antioxidant properties, most likely due to its contents in total phenols (446 mg/100 g fresh weight), total flavonoids (32 mg/100 g fresh weight), vitamin C (29.9 mg/100 g fresh weight), and β-carotene (1.062 mg/100 g fresh weight) [36]. When it comes to the analyses of the University of Thessaly, dry *B. edulis* has been shown to contain 347 kcal/100 g, 21.9 g/100 g proteins, 59.2 g/100 g carbohydrates, 2.6 g/100 g fat, and 6.4 g/100 g ash. In addition, it is worth mentioning that toxic metallic elements have been found in Boletus edulis, such as Cd and Pb, but at concentrations that are considered harmless [35]. Recently, *B. edulis* has been shown to possess anti-tumor properties. One study isolated a new protein from the dried fruit bodies of *B. edulis*, which in vitro exhibited potent anti-cancer activity on A549 cells, and this cytotoxicity was mediated by the induction of apoptosis and arrest of A549 cells in the G1 phase of the cell cycle [37]. Another study isolated a novel cold-water-soluble polysaccharide from *B. edulis*, which induced apoptosis of MDA-MB-231 and Ca761 cells through cell block in the S phase and G0/G1 phase, respectively [38]. Lastly, an issue regarding significant amounts of nicotine in dried wild mushrooms (mainly *B. edulis* from China) was reported to the European Commission which resulted in the European Food Safety Authority (EFSA) proposing temporary maximum residue levels of 0.036 mg/kg for fresh wild mushrooms and 1.17 mg/kg for dried wild mushrooms (2.3 mg/kg for dried ceps only) [39].

### 4.5. Cantharellus cibarius

*Cantharellus cibarius* (*C. cibarius*), known as yellow chanterelle, is an ectomycorrhizal mushroom that belongs to the Cantharellaceae family and grows in Asia, America, Africa, and Europe [40]. *Cantharellus cibarius* contains protein up to 53.7 g/100 g DM, 31.9 g/100 g DM carbohydrates, and is one of the few mushrooms that has a higher SFA content (926.953 mg/kg DM) than PUFA (655.176 mg/kg DM) or MUFA (148.493 mg/kg DM) [40,41]. According to the analyses of the University of Thessaly, the protein content was found to be 19.9 g/100 g DM, the carbohydrate content was found to be 43.5 g/100 g DM, and exhibited a high SFA content, at 42.3% of the total fat content. Regarding phenolic compounds, *C. cibarius* has been shown to contain protocatechuic acid (42.79 µg/g DM), p-hydroxybenzoic acid (15.68 µg/g DM), caffeic acid (16.34 µg/g DM), ferulic acid (10.38 µg/g DM), gallic acid (161.83 µg/g DM), homogentisic acid (316.76 µg/g DM), pyrogallol (91.09 µg/g DM), myricetin (23.37 µg/g DM), and catechin (5.82 µg/g DM) [42]. Moreover, *C. cibarius* exhibits biological properties, such as antioxidant, immunomodulatory, anti-inflammatory, anti-viral, and anti-microbial properties [40]. In addition, polysaccharides from *C. cibarius* have been shown to exhibit an anti-tumor effect in human colon cancer cells LS180 through perturbation in the G0/G1 and S phases of the cell cycle, as well as through the attenuation of activated nuclear factor kappa B (NF-κB) phosphorylation and inhibition of IκBα (nuclear factor of kappa light polypeptide gene enhancer in B-cells inhibitor, alpha) [43].

### 4.6. Coprinus comatus

*Coprinus comatus* (*C. comatus)* belongs to the family Agaricaceae and is also known as shaggy mane, chicken drumstick mushroom, and lawyer's wig [44]. *Coprinus comatus*'s cap is usually white, but over time it turns pink and covers the stipe over [45]. When it comes

to the nutritional value, *C. comatus* contains 368–525 kcal/ 100 g DM, 49.2–76.3 g/100 g DM carbohydrates, 11.8–29.5 g/100 g DM protein (with glutamic acid and alanine being the amino acids with the largest concentration), 1.1–5.4 g/100 g DM fat, 66% of which are polyunsaturated fatty acids (PUFA) [44], and the remaining nutrients were found to be in similar percentages in the with analyses of the University of Thessaly except from the fat (14.2% DM protein, 53.8% DM carbohydrates and 0.9% DM fat). When it comes to macroelements, *Coprinus comatus* contains phosphorus (5.726 mg/kg DM), potassium (4.077 mg/kg DM), magnesium (1.348 mg/kg DM), sodium (291.7 mg/kg DM), and calcium (157.2 mg/kg DM). The major microelement content consists of iron (237.9 mg/kg DM), zinc (53.25 mg/kg DM), and manganese (10.97 mg/kg DM) [46]. Furthermore, the most prominent phenolic compounds that have been detected in *Coprinus comatus* are p-hydroxybenzoic acid (11.73 µg/ g DM), p-coumaric acid (8.86 µg/g DM), and cinnamic acid (4.07 µg/g DM) [8]. *Coprinus comatus* possesses several biological properties, the most prominent of which, is its antidiabetic properties [47–49]. Furthermore, it possesses antioxidant [50], anti-inflammatory [51], hepatoprotective [52], anti-cancer [53], and anti-microbial properties [54]. Moreover, *C. comatus* has been shown in vivo to possess hypoglycemic effects through the inhibition of $\alpha$-amylase activity, which is an enzyme that plays a major role in the digestion of starch and hydrolyzes $\alpha$-1,4 glycosidic bonds [50]. According to European Food Safety Authority (EFSA) recommendations, consumption of *C. comatus* can provide 10% of zinc RDI [8].

### 4.7. Cordyceps militaris

*Cordyceps militaris (C. militaris)*, also known as caterpillar fungus (Ascomycota), is not included in the European documents [55], but is widely consumed as a health food and is used in traditional medicines in China and South East [56]. In order to have health claims, the conditions of use of 400–800 mg/day are necessary and it could be antioxidant due to polysaccharides content but on the basis of the data presented, the Panel concludes that a cause and effect relationship has not been established between the consumption of the *C. militaris* which are the subject of this opinion and antioxidant properties [57]. *Cordyceps militaris* may also be beneficial against chronic kidney disease by affecting the toll-like receptor 4/nuclear factor-kappa B (TLR4/NF-κB) signaling pathway, as well as by improving the lipid profile and redox capacity in chronic kidney disease patients [58]. The aforementioned effects may be owed to cordycepin, which is a purine nucleoside antimetabolite and antibiotic isolated from *Cordyceps militaris* [58].

### 4.8. Craterellus cornucopioides

*Craterellus cornucopiodes* (*C. cornucopioides*), or horn of plenty or black trumpet, belongs to the family Cantharellaceae and is an edible fungus that is also considered a source of valuable bioactive compounds. Regarding its proximate composition, *C. cornucopioides* contains 21.04 g/100 g DM protein, 64.72 g/100 g DM carbohydrates, 5.87 g/100 g DM fat, 8.37 g/100 g DM ash, and 1726.11 kJ/100 g DM energy [59]. According to the analyses of the University of Thessaly, the protein content was found to be 19.5 g/100 g DM, the carbohydrate content was found to be 45.7 g/100 g DM, and exhibited a high SFA content, at 69.6% of the total fat content. When it comes to bioactive compounds, it contains quercetin (39.64 µg/g DM), p-coumaric acids (8.65 µg/g DM), caffeic acid (16.2 µg/g DM), gallic acid (0.74 µg/g DM), ascorbic acid (0.81 mg/g DM), ergosterol (3.27 mg/g DM), $\alpha$-tocopherol (1.15 µg/g DM), $\gamma$-tocopherol (0.62 µg/g DM), $\delta$-tocopherol (0.17 µg/g DM), and possesses antioxidant properties [59]. In addition, the effects of a novel *C. cornucopioides* polysaccharide on immunosuppressive BALB/c mice was studied and the results showed significant increases in spleen and thymus weight indices and that the polysaccharide could upregulate the protein expression of the G-protein-coupled cell membrane receptor TLR4 and the production of its downstream protein kinases (TRAF6, TK1, p-IKKα/β, and NF-κB p50), thus exhibiting immunomodulatory effects [60].

### 4.9. Craterellus lutescens

*Cantharellus lutescens* belongs to the family Cantharellaceae, is also known as Yellow Foot, and has been found to exhibit cytotoxicity against human cancer strains and inhibition of nitric oxide (NO) production, as well as weak antimicrobial activity against *Candida albicans* [61]. A research study determined the protein content, fiber content and total free amino acids and they were found to be 25.07%, 15.87%, and 17.87 mg/g (d. w.), respectively [41]. Different percentages emerged from the analyses of the University of Thessaly and specifically protein (14.5%), carbohydrates (52.4%), and fat (5.5%). Although it is one of the most widely cultivated mushrooms in Spain, it has no health claims from EFSA due to the fact that there are only a few dietary surveys in Europe.

### 4.10. Ganoderma lucidum

*Ganoderma lucidum* (*G. lucidum*) belongs to the family Ganodermataceae and is most popular for its medicinal importance rather than its nutritional benefits since it has a hard texture and bitter taste [62]. It has been used in traditional Chinese and Japanese medicine as a herbal remedy for hundreds of years [63]. Regarding its composition, it has been found that *G. lucidum* consists of 44.95% carbohydrate, 15.75% protein, 14.81% crude fiber, 12.99% moisture, and 4% ash [62]. When it comes to the analyses of the University of Thessaly, the most important outcome is not the percentage of protein (19.2%), carbohydrates (57.85), and fat (2.1%), but the high percentage of unsaturated fat which reaches 78.3% of the total fat content. More than 279 bioactive compounds have been identified in *G. lucidum*, including terpenoids (meroterpenoids, lucidenic acids, ganoderic acids) and polysaccharides ($\beta$-D-glucans, $\alpha$-D-glucans, $\alpha$-D-mannans) [63–65]. Regarding its pharmacological value and biological properties, *G. lucidum* has exhibited antioxidant [66], anti-diabetic [67], anti-cancer [68], anti-inflammatory [69], and cardioprotective properties [70]. Furthermore, polysaccharides from *G. lucidum* attenuate the production of pro-inflammatory cytokines and could have neuroprotective effects [69]. In addition, *G. lucidum* polysaccharides possess immune-modulating effects through the activation and the expression of cytokines associated with inflammatory response (such as interleukin-1, interleukin-6, and tumor necrosis factor-$\alpha$) or anti-tumor activity (such as interferon-$\gamma$ and tumor necrosis factor-$\alpha$) [68]. According to EFSA, 150–350 mg daily intake powder helps in the reduction of cholesterol levels and immunity. Lastly, *G. lucidum* presents a unique and scarce combination: classified as a medicinal mushroom and as an adaptogen herb, which enhances the body's resistance to many environmental, chemical, and biochemical factors at no cost to the operation [71].

### 4.11. Grifola frondosa

*Grifola frondosa* (*G. frondosa*) belongs to the family Grifolaceae and is known as cloud mushroom, sheep's head, and king of mushrooms. It is cultivated in Asia, Europe, and North America and is highly regarded for its taste, nutritional and pharmacological value [72]. *Grifola frondosa* is made up of 70–80 g/100 g DM carbohydrates, 13–21 g/100 g DM protein, 1.5–6.5 g/100 g DM fat, and 4.8–7.1 g/100 g DM ash [72]. Except for the fat which is higher (2.6%), protein (13.8%) and carbohydrates (61.3%) are similar to the values found in the analyses conducted at the University of Thessaly. When it comes to bioactive compounds, *G. frondosa* contains 3.8% water-soluble polysaccharides, 13.2% of which is $\beta$-D-glucan [73]. The aforementioned polysaccharides appear to be of major importance regarding the medicinal and biological properties of *G. frondosa*, which include anti-cancer [74,75], anti-diabetic [76], hypolipidemic [77], and antioxidant properties [78]. Regarding the anti-cancer effects of *G. frondosa*, a recent in vivo study demonstrated that a novel acid-soluble polysaccharide isolated from *G. frondosa* could protect thymuses and spleens of tumor-bearing mice and inhibit the growth of H22 solid tumors, as well as significantly improve the activities of NK cells, macrophages, CD19+ B cells, and CD4+ T cells, leading to the apoptosis of H22 cells via G0/G1 phase arrested [75]. Last but not least, *G. frondosa* related to the following claimed effect: blood glucose control [79].

### 4.12. Hericium erinaceus

*Hericium erinaceus* (*H. erinaceus*) belongs to the family Hericiaceae, is also known as lion's mane, monkey's head mushroom, and Yamabushitake, is a mushroom that is widely found in East Asian countries, and is considered to possess significant medicinal value [80]. Specifically, it has been shown to possess antioxidant [81], anti-inflammatory [82], anti-diabetic [83] and anti-cancer properties [84]. Lately, research has been focused on the positive effects of *H. erinaceus* on brain health and antidepressant-like effects. A clinical study in 77 volunteers with a body mass index > 25 kg/m$^2$ found that *H. erinaceus* significantly reduced depression and anxiety, as well as improved sleep disorders after 8 weeks of oral administration [85]. A recent review concluded that *H. erinaceus* may significantly ameliorate depressive disorder, probably due to bioactive compounds, such as hericenones (aromatic compounds) and erinacines (erinacines belong to a group of cyathin diterpenoids), which are known to contribute to antidepressant-like effects [1]. A potential mechanism for the anti-depressant effects of *H. erinaceus* includes the stimulation of neurotrophic factors, such as nerve growth factor (NGF), which has been shown to be associated with neurogenesis and neuroplasticity [1]. In the analyses of the University of Thessaly, the carbohydrate content was found to be 59.2%, the protein content was 19.9%, and the fat content was 3.6%. The same percentages emerged from the analyses of other researchers [86].

### 4.13. Lentinula edodes

*Lentinula edodes* (*L. edodes*) belongs to the family Omphalotaceae, is also known as Shiitake, is cultivated in Europe, Asia, Australia, and North America accounting for 17% of the global edible fungi supply, and is the second most popular edible mushroom worldwide [87]. *Lentinula edodes* has been found to consist of 58–60% carbohydrates, 20–23% protein, 9–10% fiber, 3–4% fat, and 4–5% ash [87–89]. Except for the fiber (1.3%) and fat (1.3%) percentage, which is lower, the remaining nutrients were found to be in similar percentages in the analyses of the University of Thessaly. Furthermore, *L. edodes* has been shown to contain high concentrations of calcium, iron, phosphorus, potassium, zinc, and manganese [90]. In addition, it contains several bioactive compounds with great nutritional and pharmacological value, such as polysaccharides, glycoproteins, and phenolic compounds (p-hydroxybenzoic, p-coumaric, and cinnamic acids) [90]. Regarding the beneficial effects of *L. edodes*, several studies have revealed its antioxidant [91], anti-cancer [92], anti-inflammatory [93], and immunomodulatory properties [94]. In addition, a randomized controlled trial with fifty-two healthy adults showed that eating dried *L. edodes* for 4 weeks (5–10 g/day) may improve immunity by increasing the proliferation of γδ-T cells and natural killer T (NK-T) cells, while also reducing serum C-reactive protein (CRP) levels [95]. Lastly, it has been previously used in order to develop sweet and salty cereal bars with enhanced nutritional and functional properties [90]. It also contributes to natural immunological defenses according to EFSA and helps support the body's immune system, by boosting the immune system and increasing the level of some immunocytes [96].

### 4.14. Marasmius oreades

*Marasmius oreades* (*M. oreades*) belongs to the family Marasmiaceae, is also known as the fairy ring mushroom, and grows in ringlike forms in grassy areas, such as lawns and meadows. *Marasmius oreades* has been shown to contain phenolic compounds (10.990 mg GAE/100 g DM) and flavonoids (1.139 mg querceting equivalent/100 g DM), with the main phenolic compounds being catechin, ferulic acid, gallic acid, and vanillic acid. In addition, *M. oreades* has been found to exhibit antioxidant effects [97]. Specifically, the antioxidant capacity of *M. oreades* ethanol extract was determined using the DPPH assay and the results showed that the *M. oreades* ethanol extract scavenged approximately 80% of DPPH free radicals [97]. Additionally, when it comes to the analyses of the University of Thessaly *M. oreades* was found to contain 38.5 g/100 g DM protein, 36.1 g/100 g DM carbohydrates, 3.8 g/100 g DM fat, 1.3 g/100 g DM ash, and 355 Kcal/100 g DM energy.

Lastly, Marekov et al. [98] evaluated the same percentage of fat (4 g/100 g DM) of Bulgarian mushrooms from *Marasmius*. Those authors reported the highest concentration of total fats in Europe.

### 4.15. Morchella elata

*Morchella elata* (*M. elata*) belongs to the family Morchellaceae, is also known as black morels, is a rare mycorrhizal fungus originating with pines and is one of the few species that prefer to grow on burnt areas [99]. When it comes to the mineral content, *M. elata* has been shown to contain K (21 mg/g), P (16.9 mg/g), S (6.57 mg/g), Ca (1039.59 µg/g), Cu (29.49 µg/g), Zn (150.32 µg/g), Na (668.3 µg/g) and Mg (870 µg/g) [99]. *M. Elata* has been shown in the analyses of the University of Thessaly to contain 28.2% protein, 33.5% carbohydrates, 13.4% fiber, and 3.6% fat, which provides similar proportions of macronutrients as the U.S. Dietary Reference Intakes recommend [100]. Lastly, *M. elata* possesses antioxidant properties and has a total phenol content of 1.732 mg gallic acid equivalent (GAE)/g extract and 0.46 mg GAE/g dried mushroom [99].

### 4.16. Pleurotus citrinopileatus

*Pleurotus citrinopileatus* (*P. citrinopileatus*), also known as golden mushroom, belongs to the family Pleurotaceae and is a highly nutritious mushroom that contains proteins, amino acids, polysaccharides, and other bioactive compounds [101]. Apart from its culinary value, *P. citrinopileatus* has a high medicinal value and has been reported to possess antioxidant, anti-inflammatory, immunomodulatory, anti-cancer, and anti-hypertensive properties [101]. In addition, *P. citrinipileatus* has exhibited hypolipidemic and anti-obesity effects. A recent study investigated the effects of *P. citrinopileatus* water extract in high-fat diet-induced obese C57BL/6J mice and the results showed that *P. citrinopileatus* significantly reduced weight gain, fat accumulation, food intake, and also decreased serum triglycerides, cholesterol, low-density lipoprotein, aspartate transaminase, creatinine and improved glucose tolerance [102]. The chemical composition of five mushroom species was determined. Some of these species have been less studied such as *Pleurotus citrinopileatus var. cornucopiae*, *P. salmoneo stramineus*, or *Pholiota nameko*. The findings showed that protein, sugar, and fat contents ranged from 16.2 to 26.6, 52.7 to 64.9, and 2.3 to 3.5 g/100 g dry mushroom, respectively [86]. In the analyses of the University of Thessaly, *P. citrinopileatus'* protein, carbohydrates, and fat content is 37.6%, 36.3%, and 2.2%, respectively, while there was a higher content in MUFA and PUFA than SFA.

## 5. Nutritional Value

Mushrooms are a good source of proteins and amino acids apart from vitamins and minerals. Their protein content varies from 14–39% on a dry weight basis as we see from the analysis below. Mushroom proteins contain most of the essential amino acids. They are also rich in carbohydrates. Total carbohydrates of dry mushrooms were also found to vary from 61.34% (*Grifola frondosa*) to 32% (*Amanita caesarea*) on the dry wt. basis. It is mainly composed of mannitol, glycogen, and hemicellulose together with a smaller amount of reducing sugars. Mushrooms are rich in different kinds of vitamins and minerals which are absent in several vegetables and meat. Drying, especially when high temperatures are applied, can cause the degradation of polysaccharides, proteins, and flavor compounds. Freezing is one of the best methods to extend mushrooms' shelf life but causes the loss of vitamins. Edible coatings and films improve the total sugar, ascorbic acid, and bioactive compounds preservation during the storage period [103]. Nutritional values of some commonly consumed mushrooms are given below (Tables 1–3). Mushrooms have a strong capacity to absorb potentially toxic trace elements from soils, including mercury (Hg), lead (Pb), cadmium (Cd), and arsenic (As), accumulate them in their bodies, and their concentrations in mushrooms can exceed the levels as found in other papers [104].

**Table 1.** Proximate analysis of edible mushrooms dry weight basis percent (TEI and University of Thessaly).

| Mushroom Species | Ash | Energy | Protein | Sugar | Carbohydrates | Fibres | Fat | Saturated | Unsatu-Rated | Salt |
|---|---|---|---|---|---|---|---|---|---|---|
| *Agaricus bisporus* | 9.3 | 336 Kcal/100 g–1424 Kj/100 g | 25.1 | <0.1 | 52.7 | 2.9 | 1.4 | 23.9 | 76.1 | <0.1 |
| *Agaricus Blazei* | 8.4 | 335 Kcal/100 g–1418 Kj/100 g | 28.6 | <0.1 | 48.4 | 6.0 | 1.6 | 41.8 | 58.2 | 1.0 |
| *Amanita caesarea* | 14.4 | 304 Kcal/100 g–1276 Kj/100 g | 24.0 | <0.1 | 31.9 | 14.9 | 5.6 | 29.9 | 70.1 | 1.1 |
| *Boletus edulis* | 6.4 | 347 Kcal/100 g–1453 Kj/100 g | 21.9 ± 0.2 | <0.1 | 59.2 | 9.9 | 2.6 | 26.2 | 73.8 | <0.1 |
| *Cantharellus cibarius* | 13.2 | 298 Kcal/100 g–1257 Kj/100 g | 19.9 | <0.1 | 43.5 | 12.4 | 2.2 | 42.3 | 57.8 | <0.1 |
| *Coprinus comatus* | 10.5 | 298 Kcal/100 g–1260 Kj/100 g | 14.2 | <0.1 | 53.8 | 12.3 | 0.9 | 24.3 | 75.7 | <0.1 |
| *Cordyceps militaris* | 4.4 | 317 Kcal/100 g–1341 Kj/100 g | 23.1 | 0.9 | 49.3 | 11.9 | 0.4 | 34.7 | 65.5 | 0.3 |
| *Craterellus cornucopioides* | 13.3 | 329 Kcal/100 g–1387 Kj/100 g | 19.5 | <0.1 | 45.7 | 7.7 | 5.9 | 30.4 | 69.6 | 0.1 |
| *Craterellus lutescens* | 8.9 | 342 Kcal/100 g–1440 Kj/100 g | 14.5 | <0.1 | 52.4 | 12.3 | 5.5 | 44.0 | 55.4 | 0.5 |
| *Ganoderma lucidum* | 2.8 | 367 Kcal/100 g–1553 Kj/100 g | 19.2 | <0.1 | 57.8 | 11.3 | 2.1 | 21.7 | 78.3 | <0.1 |
| *Grifola frondosa* | 1.5 | 346 Kcal/100 g–1462 Kj/100 g | 13.8 | 0.3 | 61.3 | 11.4 | 2.6 | 22.9 | 77.1 | 0.1 |
| *Hericium erinaceus* | 7.1 | 355 Kcal/100 g–1502 Kj/100 g | 19.9 | <0.1 | 59.2 | 3.3 | 3.6 ± 0.1 | 41.8 | 58.2 | <0.1 |
| *Lentinula edodes* | 6.1 | 340 Kcal/100 g–1443 Kj/100 g | 20.7 | <0.1 | 59.5 | 3.8 | 1.3 | 25.3 | 74.7 | <0.1 |
| *Marasmius oreades* | 1.3 | 355 Kcal/100 g–1500 Kj/100 g | 38.5 | <0.1 | 36.1 | 11.6 | 3.8 | 36.0 | 64.0 | <0.1 |
| *Morchella elata* | 11.5 | 306 Kcal/100 g–1291 Kj/100 g | 28.2 | <0.1 | 33.5 | 13.4 | 3.6 | 12.8 | 87.3 | <0.1 |
| *Pleurotus citrinopileatus* | 7.9 | 330 Kcal/100 g–1395 Kj/100 g | 37.6 | <0.1 | 36.3 | 7.0 | 2.2 | 42.1 | 57.9 | <0.1 |

**Table 2.** Nutritional value of mushrooms (vitamins per 100 g DM).

| Mushroom Species | A | B1 | B2 | B12 | B5 | B6 | B3 | C | D |
|---|---|---|---|---|---|---|---|---|---|
| *Agaricus bisporus* | 0 [105] | 0.88–1.2 [105] | 5.3–6.4 [105] | 0.00053–0.0013 [105] | 1.7 [105] | 1.1 [105] | 36–57 [106] | 27.7 [105] | 0 [107] |
| *Agaricus Blazei* | 0.001 [108] | 1.21 [108] | 3.41 [108] | 0.0017 [108] | 39.4 [108] | 0.83 [108] | 39.9 [108] | 12.65 [108] | 0.018 [108] |
| *Amanita caesarea* | | 0.02–1.6 [106] | 0.3–4.5 [106] | | | | 1.3–2.7 [106] | 207 [109] | |
| *Boletus aereus* | 0.000782 [36] | 0.37 [36] | 0.82 [36] | 0.00039 [107] | | 0.006 [36] | 14.72 [36] | 9.3 [36] | 0.0047 [110] |
| *Cantharellus cibarius* | | 0.12 [111] | 0.11 [111] | 0.00208 [112] | 0.90 [111] | | 6.42 [111] | 1.96 [111] 42 [109] | |
| *Coprinus comatus* | | 0.06 [113] | 0.23 [113] | | | | 3.55 [113] | 6.8 [113] | |
| *Cordyceps militaris* | 96 [114] | | 0.16 [114] | | | | 4.9 [114] | | 0 [114] |
| *Craterellus cornucopioides* | | 0.11 [111] | 0.06 [111] | 0.00109–0.00265 [107] | 0.86 [111] | | 3.34 [111] | 1.89 [111] 87 [109] | 0.0047 [106] |
| *Craterellus lutescens* | | | | | | | | 0.61 [61] | 0.00139 [61] |
| *Ganoderma lucidum* | | 3.49 [115] | 17.10 [115] | | | 0.71 [115] | 61.9 [115] | | |

**Table 2.** *Cont.*

| Mushroom Species | A | B1 | B2 | B12 | B5 | B6 | B3 | C | D |
|---|---|---|---|---|---|---|---|---|---|
| *Grifola frondosa* | 0 [116] | 0.15 [116] | 0.36 [116] | 0 [116] | 0.68 [116] | 0.06 [116] | 3.89 [116] | | 0.41 [116] |
| *Hericium erinaceus* | | | | 0.04–0.36 [117] 0.56 & 1.04 [117] | | | | | |
| *Lentinula edodes* | | 0.6 [118] | 1.8 [118] | 0.00561 [106] | | | 12–99 [106] | 25 [118] | 0.001 [118] |
| *Marasmius oreades* | | | | | | | 1.2–6.6 [106] | No data [119] | |
| *Morchella esculenta* | | | | 0.00012 [107] | | | | 13 [119] | |
| *Pleurotus ostreatus* | | 0.9 [118] | 2.5 [118] | 0.0006 [118] | | | 34–109 [106] | 20 [118] | 0.0003 [118] |

**Table 3.** Major essential, non-essential, and toxic element concentrations (mg/100 g on dry weight basis) in sixteen species of mushrooms.

| Mushroom Species | Fe (mg) | Zn (mg) | Mg (mg) | Se (mg) | Cu (mg) |
|---|---|---|---|---|---|
| *Agaricus bisporus* | **18.5** [120] | **55.7** [121] | **108.8** [121] | **3–5** [121] | **29.2** [121] |
| *Agaricus Blazei* | 79.63 [108] | 6.61 [108] | 100 [108] | 36 [108] | |
| *Amanita caesarea* | 16.9 [122] | 4.7–9.2 [106] | 12.52 [123] | | 2.97 [123] |
| *Boletus aereus* | 44 [124] | 7.72 [124] | 220 [124] | 1.23 [124,125] | 2.15 [124] |
| *Cantharellus cibarius* | 29.6 [126] | 6.24 [126] | 206 [126] | 0.17 [127] | 46.1 [127] |
| *Coprinus comatus* | 69 [120] | 23.1 [120] | 16 [120] | | 2.70 [123] |
| *Cordyceps militaris* | 14.4 [114] | 10 [114] | 3.414 [114] | | |
| *Craterellus cornucopioides* | 41.3 [106] | 0.61 [106] | 97.8 [106] | 0.14 [127] | 0.43 [106] |
| *Craterellus lutescens* | | | | Not detected [125] | |
| *Ganoderma lucidum* | 82.6 [128] | 0.7 [129] | 7.95 [129] | 0.72 [129] | 26 [129] |
| *Grifola frondosa* | 0.3 [128] | | | | |
| *Hericium erinaceus* | **11.200** [128] | **3.410** [128] | **75.810** [128] | **Not detected** [128] | **1.101** [128] |
| *Lentinula edodes* | **6.9** [128] | **6.710** [128] | **102.01** [128] | **Not detected** [128] | **1.101** [128] |
| *Marasmius oreades* | 30.5 [126] | **6.12** [113] | **9.549** [130] | 0.15 [125] | 0.923 [126] |
| *Morchella esculenta* | 19.5 [126] | 9.89 [126] | 181 [126] | | 6.26 [126] |
| *Pleurotus ostreatus* | **10.20** [128] | **4.600** [128] | **125.4 ± 0.001** [128] | **Not detected** [128] | **1.420** [128] |

Research showed that the nutrient compositions of different mushroom species vary by slight differences. Protein content ranges from the lowest of 13.8 mg/100 gm (*Grifola frondosa*) to a maximum of 38.5 mg/100 gm (*Marasmius oreades*). Carbohydrate content ranges from the lowest of 32 mg/100 gm (*Amanita caesarea*) to a maximum of 61.4 mg/100 gm (*Grifola frondosa*). Fat content ranges from the lowest of 0.4 mg/100 gm (*Cordyceps militaris*) to a maximum of 5.9 mg/100 gm (*Cordyceps militaris* Table 1).

## 6. Discussion

It has been estimated that 2–3% of the population follows a vegetarian or vegan diet. In a market research conducted in 2018, a third (34%) of British meat-eaters reported having reduced their meat consumption over the last year (compared to 28% in 2017) [131]. As we move toward a third year of the COVID-19 pandemic, diet patterns include sources of healthy fats and good choices of protein such as mushrooms except from a plant-based diet. The researchers of the University of Thessaly were motivated in order to study a food as an alternative source of protein with improved health benefits such as mushrooms.

Research on nutritional value adds to the growing list of potential health benefits of eating mushrooms. That was the reason why the Meteora Museum currently has about 70 different mushroom and truffle products. Many of the mushroom products of the Meteora Museum were developed with a rational design in order to increase their nutritional value without discounting the organoleptic character. The unique taste of some of them but also their nutritional value is ensured by the selective utilization of raw materials and the observance of traditional recipes in small-scale production.

In this context, a package of new foods was developed utilizing different types of mushrooms that are included in the pharmaceuticals but also a series of low glycemic index products such as wholemeal pasta with mushrooms and high bio-function nuts as well as spoonless desserts from different types of mushrooms and sugar substitutes.

Therefore, lentinan (a beta-glucan from the shiitake mushroom) supplementation shows promise as an anti-tumor agent. In one study, patients suffering from non-small cell lung cancer, who underwent chemotherapy, either received an intramuscular injection of 4 mg/day of lentinan in addition to chemotherapy or chemotherapy only for 12 weeks. In another study, peritoneal and epithelial ovarian cancer patients were randomized to receive either two capsules 3 times/day containing 500 mg active hexose correlated compound (AHCC) or placebo throughout six cycles of chemotherapy which correlates with overall, progression-free survival [132].

Finally, the total lipid content of *Agaricus*, *Marasmius*, and *Pleurotus*, in Bulgaria, ranged from 3–9 (g/100 g) [98] from which the percentage of saturated and unsaturated is 28% and 71.8%, respectively. The study from species collected in Greece [133] ended up finding low total lipids (0.1–0.4 g/100 g) and the species *A. bisporus* from the Netherlands and Portugal showed the highest levels of omega 3 and 9 and omega 6, respectively [134].

Dried mushrooms have the highest nutritional value [135]. Mushrooms can enrich the human diet with some macronutrients providing almost 9–40% of the daily requirement for dietary fibers [136] and to this outcome is in total agreement with the analysis of University of Thessaly. The total sugars concentration found in *Agaricus bisporus* species (white and brown mushrooms) was higher than the values found in samples from Taiwan (1.75–3.13 g/100 g), results expressed in fresh weight calculated taking into account the dry matter [137].

Additionally, a serving of 100 g of most species of mushrooms covers from 15 to 30% of the daily recommendation of vitamins and trace elements. According to the literature review, it is confirmed for the mushroom species *Agaricus bisporus*, *Amanita caesarea*, *Boletus aereus*, *Lentinula edodes*, *Pleurotus ostreatus*, and *Ganoderma lucidum* for vitamins B1, B2, B3, B12, and vitamin C (Table 2) [36,115], while the species *Cantharellus cibarius* had the lowest concentration of vitamin B2 (with) [111]. One of the highest concentrations of vitamin B2 is 5.3–6.4 mg/100 g dry weight basis in the species [104,105]. *Craterellus cornucopioides* and *Coprinus comatus* species have significant amounts of vitamins B12, B3, and C [111]. All species

had a significant amount of ascorbic acid. The variation in vitamin C is 207–6.7 mg/100 gr dry weight basis with the species *Amanita caesarea* having the highest and the species *Craterellus lutescens* the lowest antioxidant activity [106,123].

With the sole exception of *Morchella esculenta*, all the other species tested had a significant amount of vitamin B12. At this point, we would like to emphasize that mushrooms are the only food of plant origin that contains vitamin B12. The variation of B12 in the species we studied is 1.04–0 mg/100 gr dry weight basis with the species *Hericium erinaceus* having the highest concentration and the species *Grifola frondosa* the lowest [115,117]. Consumption of 100 gr dehydrated product (or 1 kg of fresh mushroom) could cover the recommended daily intake for adults, although such an amount is difficult to consume daily. However, a small amount on a daily basis could prevent vitamin B12 deficiency in vegetarians [107].

As shown in Table 2, *A. bisporus* culture appears to be a good source of B-complex vitamins and niacin [118]. For these vitamins, one serving of mushrooms may contribute about 10% or more of the recommended daily allowance according to the recommendations. *A. bisporus*, on the other hand, contains very low levels of vitamin A and vitamin D. The reason for the low levels of vitamin D2 in cultivated mushrooms seems to be that the conversion of ergosterol to ergocalciferol (vitamin D2) requires sunlight (or artificial ultraviolet light) [135]. Studies have shown that the concentration of vitamin D2 in *A. bisporus* can be increased by 467% with UV radiation after harvest. A total of 15 human trials reported on the consumption of *A. bisporus* mushrooms and physical health outcomes [138]. Research conducted among the Dutch population confirmed the positive effect of vitamin D supplementation in cases of chronic diseases, assessed on the basis of lowering mortality. A reduction in mortality was obtained in: cancers by 25%, cardiovascular diseases by 25%, diabetes by 15%, and multiple sclerosis by 50% [139]. Finnish studies monitoring the effects of supplementation with vitamin D in the amount of 50 µg/D (2000 IU) (from birth) data analysis over 31 years show a reduction in the risk of developing type 1 diabetes by 80% [140]. Infants of mothers supplementing with vitamin D3 had a significantly lower risk of dental caries, respiratory infections, and sepsis [141].

Mushrooms of the *B. edulis* species prepared for consumption met 7–14% RDA of vitamin B1 for healthy adults and 1% RDA for B2, B3, and B3, respectively [36]. Finally, mushrooms of the species *Cantharellus cibarius*, *Craterellus cornucopioides*, and *Lentinula edodes* cover 15% of RDA in vitamin B12 according to the findings of Watanabe et al. [107,142].

Mushrooms can enrich the human diet with certain micronutrients. The species met the required standards of recommended daily intake of B vitamins, ascorbic acid, and vitamin D (in *Boletus aereus* and *Craterellus cornucopioides*). On the other hand, they fall short of the required amount of daily intake of vitamin A [36]. Therefore, according to Table 2, which is based on new scientific data, we can claim that the species *Agaricus bisporus* has a high content of riboflavin (5.3–6.4 mg/100 g DM) and niacin (36–57 mg/100 g DM), while the species *Pleurotus ostreatus* only in niacin (34–109 mg/100 g DM) [36,106] and all mushroom species have a high content of vitamin C [121].

In terms of minerals, according to Table 3, mushrooms of the varieties *Boletus aereus*, *Cantharellus cibarius*, *Coprinus comatus*, *Craterellus cornucopioides*, and *Pleurotus ostreatus* have a high content of iron. The variety *Ganoderma lucidum* has the highest concentration of 82.6 mg/100 g dry weight basis and the variety *Grifola Frondosa* has the lowest and most insignificant amount of iron of 0.3 mg /100 g DM [128]. Regarding Zn, it is also found in high content in all categories of mushrooms with the exception of the *Craterellus cornucopioides*. With significant variations in Zn ranging from 55.7 mg to 0.61 mg/100 g DM for the varieties *Agaricus bisporus* and *Craterellus cornuco-pioides* respectively [106,121].

Finally, all mushroom species have a high content of magnesium and copper with the *Agaricus blazei* species being superior to all species having a high content even of selenium as well as *Boletus aereus* (1.23 mg/100 g DM). This claim is also confirmed by the international literature [23,126], with the variation of significant amounts of magnesium ranging from 220 mg to 3.414 mg/100 g dry weight basis for the varieties *Boletus aereus*

and *Cordyceps militaris*, respectively [114,124] and in copper 29.2–0.43 mg/100 g dry weight basis for the varieties *Agaricus bisporus* and *Craterellus cornucopioides*, respectively [106,121].

Therefore, from the data analysis reported in this review, it appears that the mushroom varieties examined can be important sources of zinc, magnesium, iron, and copper. Thus, a large part of the rural population can only survive on a diet based mainly on mushroom crops, dealing with malnutrition [131]. In addition, the consumption of these edible mushrooms in the diet could be an excellent source of iron, zinc, and other micronutrients even for groups of the population with high nutritional requirements such as pregnant women and children. Recent researchers studied the accumulation, release, and absorption of zinc and indole compounds from mycelial cultures using in vitro models, and the results indicate that mushrooms and their in vitro cultures not only synthesize and accumulate these compounds but also potentially release them into the gastrointestinal tract where they can be absorbed by the human body, which is reflected as a specific health benefit [143].

Considering the above, the literature we have referred to is the most recent (depending on the mushroom variety). In addition, the nutrients we studied are by no means toxic to humans in the amount we report but also in larger amounts as other researchers report. The concentrations of heavy metals in mushrooms grown in uninfected fertilizers are usually low. Differences in the concentrations of minerals and trace elements largely depend on the place of cultivation [35]. Therefore, we can argue that this is a product of our country that we must support not only for its nutritional value and health benefits but also because due to the existing treatment it does not pose a health risk [133].

This paper sums up the diverse beneficial health effects of Greek mushrooms to humans, as a dietary fiber, and as an important source of medicines. On the other hand, in the last decade an increasing number of people have actively joined and today there is an active "mushroom-loving" community that—simultaneously with the search and registration of mushrooms—combines recreation in forest ecosystems, gastronomy, mountain tourism, and cultural events. Although there are now hundreds of registered products in Greece that contain native edible mushrooms promoted in the trade, they do not seem to be preferred by consumers [144].

We compared the results and proved that the composition of each mushroom presented differences in order not only to bring home the importance and benefits of mushroom consumption but also to emphasize the value of the new section of the Mushroom Museum of Meteora which is something unique in Europe.

## 7. Educational Interventions and Mushroom Museum of Meteora

For all the above reasons, the University of Thessaly, with the Museum of Natural History and Mushrooms, was inaugurated with the ultimate goal of rationally designing new products utilizing different types of edible mushrooms, highlighting and increasing their nutritional value, without reducing nutritional value. The consequence of this cooperation is the creation of at least 70 final products with unique organoleptic characteristics and nutritional value which is ensured by the selective utilization of raw materials and the observance of traditional recipes in small-scale production.

Many of the Mushroom products of the Museum were developed with a rational design in order to increase their nutritional value without discounting the organoleptic character. The unique taste of some of them but their nutritional value is ensured by the selective utilization of raw materials and the observance of traditional recipes in small-scale production. In this context, a package of New Foods was developed utilizing different types of mushrooms that are included in the Pharmaceuticals but also a series of low glycemic index products which includes wholemeal pasta with mushrooms and high bio-function nuts as well as spoonless desserts and different types of mushrooms and sugar substitutes. All products have been created with the contribution of the School of Agricultural Sciences, University of Thessaly, with Dr. Olga Gortzi scientifically responsible, which is an additional certification of their quality and taste.

The Museum of Natural History of Meteora and the Museum of Mushrooms acquired a new wing, which concerns the nutritional and therapeutic value of mushrooms. To create this wing, a new technology was used, the so-called "Spatial Augment Reality". This is the first time that this innovative technology has been used in a permanent museum exhibition in Greece, while there are few reports for its use in museums abroad. Where they do not exist, the screenings are performed in 2D, while in the Museum, they are performed on 3D-sculpted mushrooms. This innovative technology is based on color pixel mapping, but also on the projection of graphics on a multi-level surface by a projection machine (multiplayer projection mapping). In this way, the new wing of the Museum, which has as its object the nutritional value of mushrooms, is highlighted through a feast of colors, which makes the museum experience unique. In order to cultivate the motivation, interest, and pleasure of the visitors through experiential actions, a 3D interactive exhibit of the human body was designed in the form of an interactive exhibit that in the simplest, most ingenious, and supervisory way presents the benefits of mushrooms in specific organs (Figure 1).

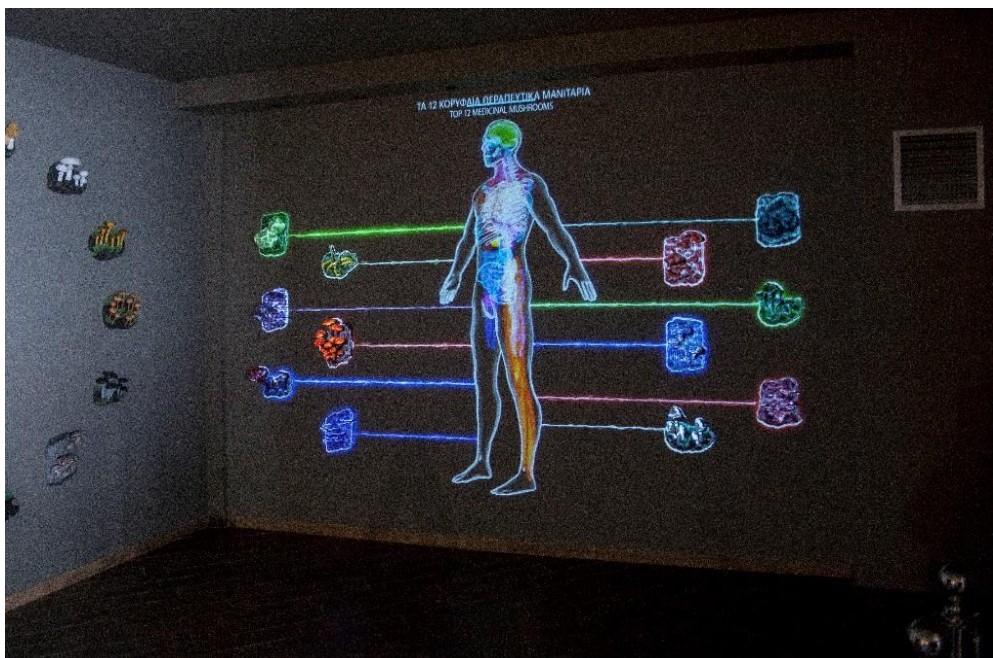

**Figure 1.** The benefits of mushrooms in specific organs.

Moreover, the new wing highlights the potential clinical and public health significance of eating mushrooms as a means of preventing disease. So, a new section, encompassing 12 top edible and 12 top therapeutic mushrooms, comes to highlight this dimension of the fungus and to supplement the educational topics of the Mushroom Museum. The choice of the 12 types of edible mushrooms was not accidental as they cover 15% of the following micronutrients: vitamin B1, B2, B3, B12, Fe, Zn, Cu, Se, and Mg, which our body needs, according to the recommended daily intake (RDI), while specific species such as *Agaricus bisporus*, *Boletus aereus*, and *Pleurotus citrinopileatus* cover 30% (Figure 2).

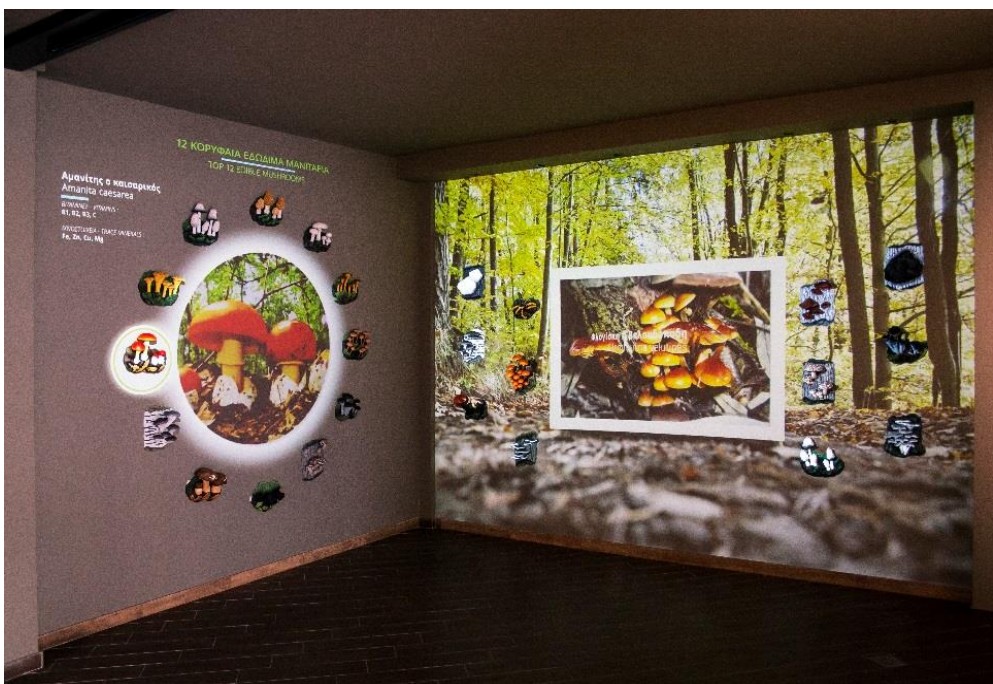

**Figure 2.** 12 top therapeutic mushrooms.

## 8. Conclusions

In conclusion, most of the mushroom species could be used as considerably good sources of macronutrients and micronutrients as well such as vitamin B, vitamin D, ascorbic acid, and minerals (Fe, Mg, Zn, Se, Cu), and their consumption can significantly contribute to the nutritional needs of people, especially in rural areas and developing countries. Regular and adequate consumption of these mushrooms can help meet the recommended daily intake of most nutrients. Knowledge of the nutritional and bioactive properties of these mushrooms will increase their conscious and safe consumption.

We could conclude about the diverse benefits of mushrooms towards humans with the words of the father of medicine, Hippocrates, "Let food be your medicine and medicine be your food". This saying aptly suits mushrooms, as they have tremendous medicinal food, drug, and mineral value, hence they are valuable assets for the welfare of humans.

A high level of fiber intake has health-protective effects and disease-reversal benefits. Persons who consume generous amounts of dietary fiber, compared to those who have minimal fiber intake, are at lower risk for developing: Cardiovascular health disease, hypertension, diabetes, obesity, and certain gastrointestinal diseases. Increasing the intake of high fiber foods or fiber supplements improves serum lipoprotein values, lowers blood pressure, improves blood glucose control for diabetic individuals, aids weight loss, and improves regularity. The current paper reflects the University of Thessaly's goal of ensuring that people benefit from the most up-to-date health and nutritional advice and highlighting the educational sections of Mushroom Meteora Museum.

## 9. Future Directions

The paper contains important information regarding the potential utilization of 16 species of mushrooms. The *Amanita caesarea, Cordyceps militaris, Craterellus lutescens, Ganoderma lucidum, Hericium erinaceus, Marasmius oreades, and Morchella esculenta* species are being studied for future research so that we can analyze their nutritional value in terms of vitamins.

**Author Contributions:** Conceptualization and methodology, O.G., A.K., O.A. and M.D. and writing—original draft preparation, S.M., M.D. and A.K.; writing—review and editing, S.M., O.A. and O.G.; supervision, O.G. All authors have read and agreed to the published version of the manuscript.

**Funding:** The authors extend their sincere gratitude to the Meteora Museum for its cooperation with the University of Thessaly and financial aid during the study.

**Institutional Review Board Statement:** Not applicable.

**Informed Consent Statement:** Not applicable.

**Data Availability Statement:** Details regarding where data supporting reported results can be found at the "Food InnovaLab" of the Department of Food Technology of Technological Educational Institute of Thessaly, Karditsa, Greece.

**Acknowledgments:** The authors would like to thank George Kostantinidis for his generous support and encouragement during the preparation this manuscript. The authors are also thankful to the University of Thessaly for providing the required facilities for this study. All authors read and approved the final manuscript.

**Conflicts of Interest:** The authors declare no conflict of interest.

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
