# Peer review of "Nutritional Composition and Biological Properties of Sixteen Edible Mushroom Species"

_applsci, doi:10.3390/app12168074_

Round 1

Reviewer 1 Report

Hello Authors,

Please see my comments below,

1) Methods of determining the nutritional values of the different mushrooms should be described.

2) Table A3: Which elements are toxic?

Author Response

  1. Methods of determining the nutritional values of the different mushrooms should be described. We agree and we added the methods on page 10
  2. Table A3: Which elements are toxic? Mushrooms accumulate toxic elements such as mercury, lead, cadmium, and arsenic and we do not include the above elements at table 3 but we added the reference before the table and thank you for the comment.

Author Response

  1. I am confused if this paper is a review or an original paper. The authors bring a review about mushrooms, but they analyzed their nutritional composition.

The abstract has been modified in order to clarify this. Specifically, “The present study aimed to briefly review the nutritional composition and biological properties of sixteen mushroom species, as well as to compare the mushrooms’ proximate composition to the analyses conducted at the University of Thessaly, Greece in cooperation with the Natural History Museum of Meteora and Mushroom Museum”.

  1. They need to include a "material and methods" section to describe the methodologies used.

A “material and methods” section has been included. Specifically: Materials and methods

A sample of dehydrated mushrooms of each variety was obtained in cooperation with the Meteora Museum. To allow greater extraction of its components, the mushroom was mashed up in a Willey type (Model ET-648, Technal Brand mill).  The physical and chemical analysis were performed at the “Food InnovaLab” Laboratory of the Department of Food Technology of Technological Educational Institute of Thessaly and at the Food Safety and Quality Control Laboratory of the School of Agricultural Sciences, of the University of Thessaly.

Chemical characterization

The whole analysis, in duplicate, has followed the official methods established by MAPA, by the Association of Official Analytical Chemists (AOAC) 100. Moisture analysis were performed using a kiln at 105°C ± 3°C for 24 hours and total ash by means of sample calcination in a muffle furnace at 550°C for 12 hours. The Kjeldahl method was utilized for protein determination, using a 6.25 correction factor. Sample fat content was detected by continuous “Soxhlet” device type extraction. Determination of total dietary fiber was based on sequential enzymatic digestion of the dried mushroom sample with alpha-amylase thermostable; protease and amyloglucosidase. The determination of carbohydrates was calculated by the difference, using rates obtained by moisture analysis, fixed mineral residue, proteins, and lipids.

  1. L30-33: Correct the English language of the sentences. Example: “Protein content ranged from the lowest 13.8 mg/100 gr to a maximum of 38.5 mg/100 g.”

The protein content of the mushrooms was found to range between 13.8 g/100 g and 38.5 g/100 g

  1. L47-48: “…from Osiris and the Chinese used mushrooms in medicine and felt that they promoted youth and health.”

This has been corrected to: “…from Osiris and the Chinese used mushrooms in medicine and felt that they promoted youth and health”.

  1. Standardize the mushrooms' presentation. If you put phylum or family for one, put it for all.

This has been standardized and the family has been included for all the mushrooms.

  1. L144: They are proximate because they are not equal. Correct this information. Never use “the same”, use “close”, “nearby” or another synonym. The results will never be equal… Correct it in the other mushrooms sections too!

When it comes to the analyses of the University of Thessaly the macronutrient content was found to be similar (protein 25.1 g/100 g DM, carbohydrate 52.7 g/100 g DM, and fat 0.9 g/100 g DM).

  1. L143-145: If these data are being published for the first time, you need to include a section "material and methods" in the paper describing the methodologies used to make it. The information in the abstract is not enough for this. Correct it in the other mushrooms sections too!

As mentioned, this has been corrected and we have modified the abstract in order to clarify this matter we have included a “material and methods” section.

  1. L220-221: “…which is nearly the same…” the same of what? Complete this sentence!

This sentence has been altered to When it comes to the analyses of the University of Thessaly the most important outcome is not the percentage of protein (19.2%), carbohydrates (57.85%) and fat (2.1%), but the high percentage of unsaturated fat which reaches 78.3% of the total fat content.

  1. L221: “... which reaches 78.3%.” 78.3% of what? Complete this sentence too!

This has been changed to “…which reaches 78.3% of the total fat content”

  1. L248-249: Here you said “analyses of the University of Thessaly”. Are the analyses not yours? Again: Whose data is this? Are these data beings published for the first time? If yes, you need to include a section "material and methods" in the paper describing the methodologies used to make it. Please, use the same person for all analyses. Correct it in the other mushrooms sections too!

As mentioned, this has been corrected and we have modified the abstract in order to clarify this matter we have included a “material and methods” section.

  1. L245: same… See sections above.

This has been corrected throughout the paper to “similar to”, “close to”, or “near”.

  1. L386: “analysis of University of Thessaly” See sections above.

As mentioned, this has been corrected and we have modified the abstract in order to clarify this matter and we have included a “material and methods” section.

  1. L387: “same” See sections above.

This has been corrected throughout the paper to “similar to”, “close to”, or “near”.

  1. L410: “our study” See sections above.

As mentioned, this has been corrected and we have modified the abstract in order to clarify this matter and we have included a “material and methods” section.

  1. L429: “our study” See sections above.

As mentioned, this has been corrected and we have modified the abstract in order to clarify this matter and we have included a “material and methods” section.

  1. L444: “our study” See sections above.

As mentioned, this has been corrected and we have modified the abstract in order to clarify this matter and we have included a “material and methods” section.

  1. L446: “same” See sections above.

This has been corrected throughout the paper to “similar to”, “close to”, or “near”.

  1. Why table A1, A2, and A3? Use only Table 1, Table 2, and Table 3.

The Tables have been changed to Table 1, Table 2 and Table 3.

  1. L556: What is AHCC? Explain here.

Active Hexose Correlated Compound (AHCC)

  1. What is “Forest mix”? Take Table 1.

This has been taken from Table 1.

21. In my opinion, some descriptions of the museum are unnecessary and take off the focus of science.
It looks like advertising in a scientific paper.
The section of the manuscript on cooperation with the Museum has been modified. The
collaboration of the University of Thessaly with the Museum is of scientific interest as a case
study of the application of new technologies in Museum education. In addition, there is scientific
interest in the development of mushroom-based functional foods.

Author Response

  1. The manuscript presents an overview of the nutritional composition and some of the biological properties of 16 common mushroom species cultivated in Greece. However, there is not clear if it is a review or an article-type manuscript. It does not have the structure of the article (e.g., Materials and Methods, Results is missing) but it shows some unpublished data (Table A1) without information about methodology, information that ensures reproducibility of the work. Instruction for authors it is stated that for 2.1.15 Review “No new, u published data should be presented. The structure can include an Abstract, Keywords, Introduction, Relevant Sections, Discussion, Conclusions, and Future Directions, with a suggested minimum word count of 4000 words.

It is referred to the abstract that the macronutrient profile of each mushroom was analyzed according to methods described in the Association of Official Analytical Chemists International and a brief review for the micronutrients it is included. The structure has been changed.

  1. The authors need to be consistent with spelling throughout the manuscript and grammar should be checked.

Thank you for your comment.

  1. The manuscript lacks in-depth discussion and further directions in conclusion section. The authors should look forward to and compare results rather than simply enumerate the literature to show some of the data already shown in the literature.

We compared the results and proved that the composition of each mushroom presented differences in order not only to bring home the importance and benefits of mushroom consumption but also to emphasize the value of the new section of Meteora Museum which is something unique in the world. Further directions in the conclusion section added.

  1. Title “Nutritional Composition and Biological Properties of sixteen edible mushroom species in Greece” should be changed as information from the manuscript cover the mushroom's a composition/properties from all over the world, not only from Greece.

The title has been changed.

  1. Lines 25/26 Re-phrase “Furthermore, the present study aimed to review the macronutrient and micronutrient composition of Greek mushrooms.” Please, check the previous comment.

We agree and it has been corrected: Furthermore, the study aimed to present the macronutrient and review the micronutrient composition not only of the more cultivated mushrooms in the world but also with the best nutrient profile.

  1. Lines 25/26 Probably “macronutrients” “micronutrients” We agree and it has been changed.
  2. Lines 26-27 Re-phrase “Indeed, medicinal mushrooms (MMs) have been studied by an interdisciplinary field of science, accompanied by an increasing number of relative studies.

Indeed, edible mushrooms have been studied for their medicinal and pharmacological effect, accompanied by an increasing number of relative studies.

  1. Information about the global mushroom market is from 2016. Why the authors do not use up-to-date information?

Thanks for the comments but we did not find up-to-date information, because the literature is quite limited and there is no recent data

  1. Line 210 “C.Comatus” should be changed to C. comatus.

We agree and it has been changed.

  1. Lines 227-228 Replace 42 from the phrase “Furthermore, polysaccharides from G. lucidum may exhibit neuroprotective effects through the down regulation of lipopolysaccharide or amyloid beta42-induced pro-inflammatory cytokines”

We agree and we changed it.

Furthermore, polysaccharides from G. lucidum attenuate the production of pro-inflammatory cytokines and could have neuroprotective effects.

  1. Line 233 “protein” not “It is source of Protein, low sodium, low fat and high fiber.“

We agree and it has been changed.

  1. Lines 234-236 The word “helps” three times in the sentence “According to EFSA 150-350mg daily intake powder helps in the reduction of cholesterol levels, help stimulate the body and it helps the immune system.”

We agree and it has been changed: According to EFSA 150-350mg daily intake powder helps in the reduction of cholesterol levels and the immunity.

  1. Lines 247-249 Add reference for this statement “Except for the fiber (1.3%) and fat (1.3%) percentage, which is smaller, all the others come in full agreement with analyses of the University of Thessaly.”

The physical and chemical analysis, were performed at the Physical Chemistry Laboratory of the Food Research Center, School of Physical Education, Sport Science and Dietetics and the Laboratory of Food Biochemistry, School of Agricultural Sciences, both from University of Thessaly from March to June 2019.  (page 10)

  1. Lines 273-275 Add reference for this statement “In our study it was found to contain 335kcal/100 g DM, 48.4 g/100 g DM carbohydrates, 28.6 g/100 g DMproteins, 1.6 g/100 g DM fat and 8.4 g/100 g DM ash, which is roughly the same.”

There is primary data shown in this manuscript

  1. Lines 303-305 Add reference for this statement “Except for the fat which is just a little higher (2.6%), protein (13.8%) and carbohydrates (61.3%) are the same as in the analysis conducted at the University of Thessaly.”

There is primary data shown in this manuscript

  1. Lines 364-365 In our study carbohydrate concentrate is 59.2%, 19.9% protein concentration. There is primary data shown in this manuscript
  2. Line 364 and 3.6% fat.

There is primary data shown in this manuscript

  1. Line 373 “M. elata” not “Concerning the macronutrients, M. Elata has been shown in our study to contain 28.2% protein, 33.5% carbohydrates, and 13.4%fibre, which is the highest percent, and 3.6% fat.”

We agree and it has been changed.

  1. Lines 531-535 Re-phrase “This need in combination with the huge volume of scientific data on the medicinal action of mushrooms motivated the team of the University of Thessaly for bibliographic review and thorough research effort in order to study as an alternative source of protein of such a cultivated food product and with improved health benefits according to EFSA.“

We agree and it has been changed : The researchers of University of Thessaly motivated in order to study a food as an alternative source of protein and with improved health benefits such as mushrooms.

  1. Lines 558-559Re-phrase “Finally, total lipid contact of Agaricus, Marasmius and Pleurotus, at Bulgaria range from 3-9 (g/100 g) according to Marekov et al. [92]

We agree and it has been changed: Finally, total lipid contact of Agaricus, Marasmius and Pleurotus range from 3-9 (g/100 g) from which the percentage of saturated and unsaturated is 28 and 71.8 respectively.

  1. and the percentage of saturated and unsaturated is 28% and 71.8%respectively.”

Finally, total lipid contact of Agaricus, Marasmius and Pleurotus range from 3-9 (g/100 g) from which the percentage of saturated and unsaturated is 28 and 71.8 respectively.

  1. Lines 560-561 It is mentioned only one study “All the studies in different European counties end up finding low total lipid (0.1-0.4g/100g) from species collected in Greece [128] …” .”

We agree and it is changed: The study from species collected in Greece end up finding low total lipid (0.1-0.4g/100g) [128]

  1. Lines 573-577 Re-phrase “This claim is confirmed for the mushroom varieties Agaricus bisporus, Amanita caesarea, Boletus aereus, Lentinula edodes, Pleurotus ostreatus, and Ganoderma lucidum for vitamins B1, B2, B3, B12 and vitamin C from Table 2 with only traces were found [73, 100] while the variety Cantharellus cibarius was also lagging behind in vitamin B2 (with the lowest concentration of 0.11 mg) [112].”

According to the literature review, it is confirmed for the mushroom varieties Agaricus bisporus, Amanita caesarea, Boletus aereus, Lentinula edodes, Pleurotus ostreatus and Ganoderma lucidum for vitamins B1, B2, B3, B12 and vitamin C (Table 2)[73, 103] while the variety Cantharellus cibarius had the lowest concentration of vitamin B2 .

  1. Line 591 “of” is missing “Consumption 100gr….”

The line exactly: Consumption of 100gr dehydrated product (or 1Kg of fresh mushroom) could cover the recommended daily intake for adults, although such an amount is difficult to consume daily.

  1. Figures A1 and A2 do not have a title

We agree and it has been changed.

Reviewer 4 Report

The article (Ref. No. 1811326) sums up diverse beneficial health effects of Greek mushrooms to humans, as a dietary fiber, and an important source of medicines. Currently, mushrooms are the most important sources of functional food and medicines. In this regard, the study is certainly relevant.

 Overall, my evaluation is positive, and I believe that the manuscript can be published in APPLIED SCIENCES, but the minor revision is recommended before publication. Some specific comments are as follows:

1.    It should be indicated in the title of the article that this is a review work

2.    Please, check the numbering of the figures. Page 17: “… of mushrooms in specific organs (Figure 1). Page 18 : Fig. A1.

3.    It should be  changed the captions under the figures because in this form they are not informative.

4.    It should be carefully checked the list of references, as recommended by the ACS style guide.

5.    This paper doesn’t provide suggestions for further research.

Author Response

  1. It should be indicated in the title of the article that this is a review work

In the abstract, it is stated that each type of fungus presented in this article is analyzed based on the method adopted from the Association of Official Analytical Chemists International due to the fact that the macronutrient profile analysis of the mushrooms took place at the University of Thessaly. We changed it the abstract in order not to mislead the reader.

  1. Please, check the numbering of the figures. Page 17: “… of mushrooms in specific organs (Figure 1).Page 18 : Fig. A1.

We agree and it has been changed.

  1. 3.    It should be changed the captions under the figures because in this form they are not informative.

We agree and it has been changed.

  1. It should be carefully checked the list of references, as recommended by the ACS style guide.

Thank you for your comments, we changed it.

  1. This paper doesn’t provide suggestions for further research.

This paper provides suggestions for further research for the species Ganoderma lucidum, Marasmius oreades, Amanita caesarea, Craterellus lutescens, Morchella esculenta, Hericium Erinaceus, Cordycept militaris (page 15) and needs to analyze their nutritional value in terms of vitamins. But thank you for the comment and we copy it from page 15 to page 19 as Future Directions.

We agree and it is changed: we copy it from page 15 to page 19 as Future Directions.

This paper provides suggestions for further research for the species Ganoderma lucidum, Marasmius oreades, Amanita caesarea, Craterellus lutescens, Morchella esculenta, Hericium Erinaceus, Cordyceps militaris (page 15) and needs to analyze their nutritional value in terms of vitamins.

Reviewer 5 Report

1.       This review contains important information regarding the potential utilization of 16 types of mushrooms.

2.       In the abstract, it is stated that each type of fungus presented in this article is analyzed based on the method adopted from the Association of Official Analytical Chemists International. In fact, this manuscript is a review as stated in the last paragraph of the introduction section. Information like the sentence above can mislead the reader as if the authors were doing the analysis directly. It is better to change the explanation so that the reader can immediately know from this abstract that this manuscript is a review. And if you want to display the method in the abstract, then the library collection method and how to process the data are displayed.

3.       All Latin names of mushrooms must be italicized.

4.       The sentences on lines 132-134 don't seem to need to be bold, but it depends on the journal style.

5.       Line 141: writing trans-cinnamic acid should be ----> trans-cinnamic acid

6.       Line 168-170: It is stated that Amanita caesarea mushroom contains protein, carbohydrate, fat and polyunsaturated fatty acid that was analyzed by University researchers. This method of citation is less common. It should be stated who published the data, or if the data has not been published, additional information should be given.

7.       Pay attention to the procedure for writing the chemical name of a compound, which must be written in italic or not.

8.       Data displayed in sentences on lines 219 - 221; 247-248; 273-275; 303-305; 373-374; 386-388; 410-411, seems to be the result of the author's analysis (primary data) which correlates with the data in Tables A1. But if this is the secondary data please put the literatures cited. However, by displaying a mixture of primary data with secondary data will lead to misleading the readers. Moreover, if there is primary data shown in this manuscript, then the method for generating the data must be written down in detail. Although technically it has been stated that the method refers to the standard AOAC analysis method, but in the body of the article it is necessary to state the reference analysis method, the source of the mushroom raw material being analyzed, and etc.

9.       How to display data in Tables A1 needs to be improved. In Table A1: on the first line better to write the title of "proximate content (%)" and then no need to write the % unit in every values below.

10.   On lines 471-483: The fungus C. militaris was declared undocumented in Europe. Does that mean this fungus is not found growing in Europe? If so, do these mushrooms in Greece come from imported goods? It is important to mention the source.

11.   In Table A3: In the second row there is Ganoderma lucidem, maybe what is meant is Ganoderma lucidum. Please re-check all the Latin names of the mushrooms displayed.

12.   Explanations related to the mushroom museum at discussion should be made in a separate section.

Author Response

  1. This review contains important information regarding the potential utilization of 16 types of mushrooms. Thank you!
  2. In the abstract, it is stated that each type of fungus presented in this article is analyzed based on the method adopted from the Association of Official Analytical Chemists International. In fact, this manuscript is a review as stated in the last paragraph of the introduction section. Information like the sentence above can mislead the reader as if the authors were doing the analysis directly. It is better to change the explanation so that the reader can immediately know from this abstract that this manuscript is a review. And if you want to display the method in the abstract, then the library collection method and how to process the data are displayed.

In the abstract, it is stated that each type of fungus presented in this article is analyzed based on the method adopted from the Association of Official Analytical Chemists International due to the fact that the macronutrient profile analysis of the mushrooms took place at the University of Thessaly and we added the methods in the article page 10 and we changed the abstract.

  1. All Latin names of mushrooms must be italicized.

We agree and it has been changed

  1. The sentences on lines 132-134 don't seem to need to be bold, but it depends on the journal style.

Thank you for your comment. We agree and we corrected it.

  1. Line 141: writing trans-cinnamic acid should be ----> trans-cinnamic acid.

Thank you for your comment .We agree and we changed it.

  1. Line 168-170: It is stated that Amanita caesarea mushroom contains protein, carbohydrate, fat and polyunsaturated fatty acid that was analyzed by University researchers. This method of citation is less common. It should be stated who published the data, or if the data has not been published, additional information should be given.

We added the methods and materials (page 10).

  1. Pay attention to the procedure for writing the chemical name of a compound, which must be written in italic or not.

Thank you for your comment. We agree and we corrected it.

  1. Data displayed in sentences on lines 219 - 221; 247-248; 273-275; 303-305; 373-374; 386-388; 410-411, seems to be the result of the author's analysis (primary data) which correlates with the data in Tables A1. But if this is the secondary data please put the literature cited. However, displaying a mixture of primary data with secondary data will lead to misleading the readers. Moreover, if there is primary data shown in this manuscript, then the method for generating the data must be written down in detail. Although technically it has been stated that the method refers to the standard AOAC analysis method, but in the body of the article it is necessary to state the reference analysis method, the source of the mushroom raw material being analyzed, and etc.

We added the methods and materials (page 10).

  1. How to display data in Tables A1 needs to be improved. In Table A1: on the first line better to write the title of "proximate content (%)" and then no need to write the % unit in every value below.

Thank you for your comment We agree it has been changed.

  1. On lines 471-483: The fungus C. militaris was declared undocumented in Europe. Does that mean this fungus is not found growing in Europe? If so, do these mushrooms in Greece come from imported goods? It is important to mention the source.

We added the source Lu, Y.; Zhi, Y.; Miyakawa, T.; Tanokura, M., Metabolic profiling of natural and cultured Cordyceps by NMR spectroscopy. Sci Rep 2019, 9 (1), 7735-7735.

  1. In Table A3: In the second row there is Ganoderma lucidem, maybe what is meant is Ganoderma lucidum. Please re-check all the Latin names of the mushrooms displayed.

 Thank you for your comment We agree it has been changed.

  1. Explanations related to the mushroom museum at discussion should be made in a separate section. Thank you for your comment We added a separate section page 17.

Round 2

Reviewer 2 Report

Check all numeric values of the text. Some numbers have dots and some numbers have commas. Standardize all numbers using dots (ex.: 0.2).

It would be interesting if you put the popular name of all mushrooms and not just a few ones.

Put the mushrooms’ sections in alphabetical order. It makes easier and more practice to search for a specific mushroom in the paper.

The material and methods section must be before the 16 mushrooms description because you used the data as information in these descriptions.

Compare your nutritional information of mushrooms with the literature. You did not make it for all 16 mushrooms; make it for all.

Format all the tables for simple spacing. Put the mushrooms in alphabetical order. It makes easier and more practice to search for a specific mushroom in the Tables.

Craterellus lutescens is missing in Table 2. Complete! What is “nd”? Put a sub legend in the Table 2 to explain this.

The sub legend of the Table 3 only has to contain the significance of “nd”. If you present values with mean ± SD, you will need to make it for all tables. Correct it (put for all or remove). You no not have letter to statistical analysis here, remove of the sub legend.

Author Response

  • Check all numeric values of the text. Some numbers have dots and some numbers have commas. Standardize all numbers using dots (ex.: 0.2).

This has been corrected. All numbers have dots.

  • It would be interesting if you put the popular name of all mushrooms and not just a few ones.

We have put the popular name of all mushrooms.

  • Put the mushrooms’ sections in alphabetical order. It makes it easier and more practice to search for a specific mushroom in the paper.

This has been corrected throughout the paper. The mushrooms’ sections are in alphabetical order.

  • The material and methods section must be before the 16 mushrooms description because you used the data as information in these descriptions.

This has been corrected and the material and methods section is now before the 16 mushrooms.

  • Compare your nutritional information on mushrooms with the literature. You did not make it for all 16 mushrooms; make it for all.

This has been corrected. We compare our nutritional information of mushrooms with the literature for all 16 mushrooms.

  • Format all the tables for simple spacing. Put the mushrooms in alphabetical order. It makes easier and more practice to search for a specific mushroom in the Tables.

This has been corrected. The mushrooms are in alphabetical order in the tables.

  • Craterellus lutescens is missing in Table 2. Complete! What is “nd”? Put a sub legend in Table 2 to explain this.

This has been corrected as concern Craterellus lutescens. Although under the table 3 it has been mentioned that ND-for not detected, it has been removed from there and is clear as not detected (and not ND) in the table.

  • The sub legend of the Table 3 only has to contain the significance of “nd”. If you present values with mean ± SD, you will need to make it for all tables. Correct it (put for all or remove). You do not have a letter to statistical analysis here, remove of the sub legend.

Values mean ± SD have been removed. It has been removed from the sub legend.